# PLASMA: Partial LeAst Squares for Multiomics Analysis

**DOI:** 10.3390/cancers17020287

**Published:** 2025-01-17

**Authors:** Kyoko Yamaguchi, Salma Abdelbaky, Lianbo Yu, Christopher C. Oakes, Lynne V. Abruzzo, Kevin R. Coombes

**Affiliations:** 1Division of Hematology, Department of Internal Medicine, Ohio State University, Columbus, OH 43210, USAchristopher.oakes@osumc.edu (C.C.O.); 2Department of Biomedical Informatics, Ohio State University, Columbus, OH 43210, USA; 3Department of Pathology and Laboratory Medicine, Medical University of South Carolina, Charleston, SC 29425, USA; 4Department of Biostatistics, Data Science, and Epidemiology, School of Public Health, Georgia Cancer Center at Augusta University, Augusta, GA 30912, USA

**Keywords:** multiomics, supervised learning, overall survival, gastric cancer, esophageal cancer

## Abstract

Advances in biotechnology have led to a boom in the generation of different kinds of biological data from the same samples, an approach called “multiomics”. So, we need better methods to combine data sources. There are now many “unsupervised” methods that ignore patient outcomes. But there is still a lack of “supervised” methods that can incorporate and learn how to predict patient outcomes. To fill this need, we introduce PLASMA, the first algorithm that can combine multiomics data and predict outcomes like overall survival. PLASMA is built on existing methods that work well on single omics data sets. We trained and tested PLASMA on publicly available stomach cancer data. We validated the model on a closely related but independent cancer type. We also tested the model on a dissimilar cancer type, where it failed to make useful predictions. This result suggests that the components of the model are biologically meaningful.

## 1. Introduction

Recent years have seen the development of numerous algorithms and computational packages for the analysis of multiomics data sets [1,2,3,4,5,6,7]. As with other applications of machine learning, problems addressed by these algorithms are divided into two broad categories: unsupervised (e.g., clustering or class discovery) and supervised (including class comparison and class prediction) [8]. Advances in the area of unsupervised learning have been substantially broader and deeper than the advances in supervised learning.

One challenge when trying to integrate multiomics data sets arises from multiple levels of missing data. At a low level, individual features in some data sets may fail to be measured in individual samples. It is usually easy to deal with low-level missing data. In some cases, they can be ignored; in others, appropriate imputation methods may be able to fill in the blanks [9,10]. At a high level, some patients may have been omitted entirely from one of the omics data sets. A common, highly conservative, response is to work with the subset of patients who have “complete data”, that is, the intersection of samples used in all data sets [11]. This approach may throw away much potentially useful data. Especially for time-to-event (“survival”) analyses, where the statistical power depends on the number of events, restricting to the complete data subset may reduce the total sample size to a point where it is no longer possible for a machine learning algorithm to learn a useful prognostic model.

As of this writing, we do not know of any supervised multiomics method that can learn to predict outcomes when some samples have only been assayed on a subset of the omics data sets. In this paper, we present a novel algorithm, Partial LeAst Squares for Multiomics Analysis (PLASMA), to predict time-to-event outcomes in the presence of incomplete data. The core idea of PLASMA relies on a two-step process, built on ideas that have proved useful for analyzing related problems in single omics data sets.

Use partial least squares (PLS) with Cox regression analysis [12] to identify factors that can predict time-to-event outcomes in each individual omics data set.For each pair of omics data sets, use samples common to both data sets to train and extend models of each factor to the union of assayed samples. For this step, we use PLS with linear regression [13,14].

First, for each of the omics data sets, we apply the PLS Cox regression algorithm to the time-to-event outcome data to learn separate predictive models (indicated in red, green, and blue, respectively, in Figure 1). Each of the individual models may be incomplete, since they are not defined for patients who have not been assayed (shown in white) using that particular omics technology. In the second step, for each pair of omics data sets, we apply PLS linear regression to learn how to predict the Cox regression components from one data set using features from the other data set. This step extends (shown in pastel red, green, and blue, respectively) each of the original models, in different ways, from the intersection of samples assayed on both data sets to their union. Then we average all the different extended models (ignoring missing data) to get a single coherent model of components across all omics data sets. Assuming that this process has been applied to learn the model from a training data set, we can evaluate the final Cox regression model on both the training set and a new test set of patient samples.

Our approach is motivated, in part, by the unsupervised method of Multi-Omics Factor Analysis (MOFA) [15,16]. MOFA, like PLASMA, does not require all omics assays to have been performed on all samples. It is still able to discover class structure across omics data sets. Also like PLASMA, it starts with a standard method—Latent Factor Analysis—that is known to work well on a single omics data set. It then fits a coherent model that identifies latent factors that are common to, and able to explain the data well in, all the omics data sets under study. Our investigation (unpublished) of the factors found by MOFA suggests that, at least in some cases, it is approximately equivalent to a two-step process analogous to the one we propose in PLASMA.

### Terminology

Because of the layered nature of the PLASMA algorithm, we intend to use the following terminology to help clarify the later discussions.

The input data contains a list of *omics data sets*.Each omics data set contains measurements of multiple *features*.The first step in the algorithm uses PLS Cox regression to find a set of *components*. Each component is a linear combination of features. The components are used as predictors in a Cox proportional hazards model, which predicts the log hazard ratio as a linear combination of components.The second step in the algorithm creates a secondary layer of components. We do not give these components a separate name. They are not an item of particular focus; we view them as a way to extend the first level components to more samples by “re-interpreting” them in other omics data sets.

## 2. Methods

Our computational method is implemented in version 1.1.3 of the plasma package, which is available from the Comprehensive R Archive Network (https://CRAN.R-project.org/package=plasma, accessed on 5 December 2024).

### 2.1. Data

The results included here are based upon data generated by the TCGA Research Network. We downloaded the entire STAD and ESCA Level 3 data sets [17,18] from the FireBrowse web site (http://firebrowse.org/ [19]) on 6 August 2018. We filtered the data sets so that only the most variable, and presumably the most informative, features were retained. To summarize:From TCGA, we obtained 140 columns of clinical, demographic, and laboratory data on 436 patient samples. We removed any columns that always contained the same value, and any columns that were duplicates of other data in the set. We also removed any columns whose values were missing in more than 25% of the patients. We converted categorical variables into sets of binary variables using one-hot-encoding. We then separated the clinical data into three parts:
Outcome (overall survival)Binary covariates (75 columns)Continuous covariates (three columns)
Exome sequencing data for 430 patients with STAD were obtained as mutation allele format (MAF) files. We removed any gene that was mutated in fewer than 4% of the samples. The resulting data set contained 1329 mutated genes.Methylation data for 388 STAD patients were obtained as beta values computed by TCGA from Illumina Methylation 450K arrays. We removed any CpG site for which the standard deviation of the beta values was less than 0.25 or for which the mean was within 0.15 of the boundary values of 0 or 1. The resulting data set contained 2291 highly variable CpGs.Already normalized sequencing data on 2588 microRNAs (miRs) were obtained for 382 patients. We removed any miR for which the standard deviation of normalized expression was less than 0.10, which left 1064 miRs in the final data set.Already normalized sequencing data on 20,531 mRNAs were obtained in 411 patients. We removed any mRNA whose mean normalized expression was less than five or whose standard deviation was less than 1.25. The final data set included 1690 mRNAs.Normalized expression data from reverse phase protein arrays (RPPA) were obtained from antibodies targeting 133 proteins in 350 patients. All data were retained for further analysis.

### 2.2. Imputation

We imputed missing data for any patient sample assayed in an input data set that yielded partial incomplete data (that is, at least some meaningful data for that particular sample). The underlying issue is that the PLS models for individual omics data sets will not make predictions on a sample if even one data point is missing. As a result, if a sample is missing at least one data point in every omics data set, then it will be impossible to use that sample at all.

For a range of available methods and R packages, see the CRAN Task View on Missing Data (https://CRAN.R-project.org/view=MissingData, accessed on 5 December 2024). We also recommend the R-miss-tastic web site on missing data (https://rmisstastic.netlify.app/, accessed on 5 December 2024). Their simulations suggest that, for the purposes of producing predictive models from omics data, the imputation method is not particularly important. Because of the latter finding, we have implemented two simple imputation methods in the plasma package:meanModeImputer will replace any missing data with the mean value of the observed data if there are more than five distinct values; otherwise, it will replace missing data with the mode. This approach works for both continuous data and for binary or small categorical data.samplingImputer replaces missing values by sampling randomly from the empirical data distribution.

For both the STAD and ESCA data, we used the sampling imputer.

### 2.3. Computational Approach

The PLASMA algorithm is based on Partial Least Squares (PLS), which has been shown to be an effective method for finding components that can predict clinically interesting outcomes [13]. The workflow of the PLASMA algorithm is illustrated in Figure 1. First, for each omics data set, we apply the PLS Cox regression algorithm (plsRcox Version 1.7.7 [12]) to the time-to-event data to learn separate predictive models. Each individual omics model consists of three kinds of regression models:The plsRcoxmodel contains the coefficients of the components learned by PLS Cox regression. The number of components is determined automatically as a function of the logarithm of the number of features in the omics data set. The output of this model is a continuous prediction of “risk” for the time-to-event outcome of interest.Next, two separate models are constructed using the prediction of risk on the training data.
-The riskModel is a coxph model using continuous predicted risk as a single predictor.-The splitModel is a coxph model using a binary split of the risk (at the median) as the predictor.


Each of these models may be incomplete, since they are not defined for patients who have not been assayed using that particular omics technology. In the second step, for each pair of omics data sets, we apply the PLS linear regression algorithm (pls Version 2.8.4 [14]) to learn how to predict the Cox regression components from one data set using features from the other data set. This step extends each of the original models from the intersection of samples assayed on both data sets to their union. Third, we average all the different extended models to get a single coherent model of components across all omics data sets. We then apply the Akaike Information Criterion (AIC) to select a final model from the set of all components arising from all individual models. Assuming that this process has been applied to learn the model from a training data set, we can evaluate the final Cox regression model on both the training set and a new test set of patient samples.

All computations were performed in R version 4.4.1 (2024-06-14 ucrt) of the R Statistical Software Environment [20]. Cox proportional hazards models for survival analysis were fit using version 3.6.4 of the survival R package [21].

### 2.4. Gene Enrichment

In order to interpret the model, we converted feature identifiers into gene names. This task was straightforward for the mutation, mRNASeq, and RPPA data sets. We used annotation files downloaded from the Illumina web site (https://webdata.illumina.com/downloads/productfiles/humanmethylation450/humanmethylation450_15017482_v1-2.csv, accessed on 5 December 2024) to extract associated genes (possibly more than one per locus) from the methylation data set. Gene enrichment (pathway or annotation) analysis was performed by uploading gene lists to ToppGene (https://toppgene.cchmc.org/) [22]. Results from ToppGene were limited to 40 entries per ToppGene category.

### 2.5. Preparing the Data

To be consistent with the MOFA R package [15], all of the data sets are arranged so that patient samples are columns and assay features are rows. Each data set includes the same complete set of patients, with columns entirely composed of “NA’s” to indicate samples that were not assayed.

## 3. Results

### 3.1. Preprocessing

To illustrate the PLASMA method, we use the stomach adenocarcinoma (STAD) and esophageal cancer (ESCA) data sets from The Cancer Genome Atlas (TCGA). We filtered the data to retain the most variable features and imputed missing data for samples assayed in an input data set with partial incomplete data (See Section 2). These data were then divided into 60% for training and 40% for testing (Figure 2). Before proceeding, we confirmed that there was no significant difference in survival between the randomly assigned training and test sets (Appendix A).

### 3.2. Individual PLS Cox Regression Models

In the first step of the PLASMA algorithm, we fit PLS Cox models on each omics data set separately. On the training set, all seven contributing data sets are able to find a PLS model that can successfully separate high-risk from low-risk patients in the training data (Figure 3).

### 3.3. Single Omics Predictions on the Test Set

Next, we compared the test results of the joint omics model to the predictions made on the test data from the separate single omics models. The Kaplan−Meier plots in Appendix A show that most of the individual models exhibit poor performance on the test data.

### 3.4. Training and Testing a Unified Model of Overall Survival

The second step of the algorithm is to extend the individual omics-based models across other omics data sets. Since this step is performed for all pairs of data sets, in our case there are seven different sets of predictions of each PLS component. These different predictions are averaged. The structure of the complete models created is the same as for the separate, individual omics models. Figure 4 shows the final composite Kaplan−Meier plot using the predicted risk, split at the median value of the training data, on both the training and test data sets. The model yields a statistically significant separation of outcomes between the high- and low-risk patients in both the training (*p* = 3.99 × 10^−18^) and test (*p* = 2.73 × 10^−8^) sets.

### 3.5. Independent Validation

Because previous unsupervised analyses of the ESCA data suggest that there are two classes of esophageal cancer (one similar to stomach adenocarcinomas and one similar to head-and-neck squamous cell cancers) [17,23], we split the ESCA data into these two types and used them both to evaluate the independent predictive power of the model (Figure 5). As we expected, the model performed poorly on the squamous cell cancers (*p* = 0.576) and performed well on the adenocarcinomas *p* = 0.025).

### 3.6. Interpreting the Model

At this point, our model appears to be a complex black box. We have constructed a matrix of components, based on linear combinations of actual features in different omics data sets. These components are then combined in yet another linear model that predicts the time-to-event outcome through Cox regression. In this section, we want to explore how the individual features from different omics data sets contribute to the models. Our first step in opening the black box is to realize that not all the components discovered from the individual omics data sets survived into the final composite model. Some components were eliminated (by stepwise feature selection based on the Akaike Information Criterion) because they appeared to be nearly linearly related to components found in other omics data sets. So, we can examine the final composite model more closely. In Table 1, we list the components that were retained in the final model, along with their coefficients, hazard ratios (“exp(coef)”), and statistical significance. While the sign of a coefficient indicates whether that component increases (positive) or decreases (negative) the hazard ratio, the full interpretation of those effects will be better understood later when we look at the contributions of individual biological features to the final model.

We see that at least one component discovered from four of the five “true” omics data sets survived in the final model; both the continuous clinical data (ClinicalCont) and the protein (RPPA) components failed. By contrast, three components from the binary clinical data were retained in the final model.

### 3.7. Final (Composite) Weights

A key point to note is that the core of the PLASMA model consists of a composition of two levels of linear models. One model starts with individual omics data sets and predicts the “components” that we have learned across all data sets. The second model uses the components as predictors in a Cox proportional hazards model of time-to-event (i.e., overall survival) data. The composite of these two steps is yet another linear model. In particular, we can compute the matrix that links the features from the omics-level data sets directly to the final survival outcome. Features from the individual omics data sets with the highest weight for that omics data set in the final model are shown in Figure 6. (We omitted continuous clinical features from this figure since there are only three features, none of which were significant. Also, note that the ninth label on the “ClinicalBin” plot is left-truncated; the full version reads “histological.type.stomach..adenocarcinoma..diffuse.type”.)

### 3.8. Gene Enrichment Using ToppGene

For each component, we determined the list of significant genes (arbitrarily defined as those in the top 5% by absolute value) contributing to that component. We uploaded those gene lists to the ToppGene web site [22] to perform gene enrichment analyses. We performed an additional gene enrichment analysis using the union of all significant genes from all components. Finally, we performed a similar analysis using the genes with the largest (top 5% by standardized weight) contribution to the final composite model. We clustered both the ToppGene annotation categories and the model components based on correlation between the vectors of the negative log10 *p*-values of the enrichment scores. Clustering determined the order of the rows and columns of the pathway annotations presented in Figure 7. A more conventional heatmap display of the pathway annotations is also included as Appendix A. Corresponding bubble plots for other ToppGene database categories (biological process, cellular location, disease, and drug) are shown in Appendix A.

## 4. Discussion

One of the most interesting aspects of this study is that the PLASMA algorithm successfully discovered a model on the training set that generalized to produce statistically significant results on the test set (Figure 4). Further, it accomplished this goal even though most of the individual omics models did not generalize well to the data set (Appendix A). We can think of three possible reasons to explain this performance.

“Wisdom of the crowd”. There has long been an idea in the machine learning field that combining ensembles of weak models can give rise to a strong model. Well-established examples of this idea are bagging and boosting [24]“Out of phase”. Each omics data set may overfit the model in a different way. Instead of reinforcing each other, the extent of overfitting may cancel out.“Feature Elimination”. The combined method successfully identifies useful predictive factors. So, we are still able to fit a generalizable Cox model on the final components.More research will need to be conducted on a variety of data sets to determine whether this phenomenon is more general.

In the original analysis of the STAD dataset by the TCGA Research Network [18], the authors identified four molecularly distinct subtypes: tumors positive for Epstein–Barr virus; microsatellite unstable tumors; genomically stable tumors; and tumors with chromosomal instability. These subtypes, albeit possessing distinct molecular characteristics, were not significantly different from each other in terms of survival outcomes. We also performed an unsupervised analysis using MOFA, and found that the latent factors it uncovered were not related to survival (Appendix A). Thus, we took a different approach: we trained a multiomic model supervised on overall survival, and then dissected the feature weights of omics assays.

First, however, we confirmed that the PLASMA model for STAD could be validated in an independent cohort. We divided the TCGA-ESCA data into esophageal adenocarcinoma (EAC) and esophageal squamous cell carcinoma (ESCC). Based on known biological similarities between EAC and STAD [17,23], we hypothesized that external validation of the PLASMA STAD-model with EAC would yield positive results, and validation with ESCC would serve as a negative comparison. Indeed, we observed that the Kaplan−Meier plot of OS shows little separation between the low- and high-risk patient groups in the validation with ESCC, while there is significant separation in the validation with EAC (Figure 5).

We recognize that a prognostic or predictive model that depends on multiple genome-wide omics technologies is unlikely to move into clinical use soon. The costs to perform the assays, both in dollars and in the delays imposed by waiting for all results, limit its clinical applicability. We expect that it will be necessary to refine the model to use a much smaller set of molecular predictors before it can be clinically useful. Toward that eventual goal, we performed several analyses to determine where to focus our attention.

One approach involved performing gene enrichment analysis to understand the biological role of the complicated factors that PLASMA found across omics data sets (Figure 7). Somewhat surprisingly, the final gene set had weaker associations with any of the biological pathways than did the individual components derived from single omics datasets. However, a heatmap of the same values (Appendix A) indicates that the annotation categories for the final gene list are strongly correlated with those for the union of all the component gene sets. It is possible that the associations are weaker because the final model retains fewer related/redundant genes than the individual PLS models. Future research in other disease contexts will be required to understand this phenomenon.

Nevertheless, the gene enrichment analyses may provide a useful way to focus on important factors for overall survival in gastric cancer. The left and right ends of the bubble plot (and the heatmap) mainly contain pathways that distinguish between the factors. But the central portion of the plots, from the KEGG_ERBB_SIGNALING_PATHWAY on the left to the WP_GLIOBLASTOMA_SIGNALING_PATHWAYS on the right, primarily consists of signaling pathways that have been identified as important across all the multiomic factors we found. Moreover, many of these signaling pathways have already been identified in the literature as important in gastric cancer. These include signaling pathways involving ERBB (EGFR) [25], Gastrin [26], Leptin [27], KIT [28], Oncostatin M [29], CKAP4 [30], and the PI3K-AKT-MTOR pathway [31]. There is also a strong association with Prolactin signaling, which has previously been highlighted for its role in breast cancer [32] and in colon cancer [33].

One limitation of gene enrichment analysis is that it ignores the direction of association between pathways (or genes) and survival. Genes with both positive and negative weights, either for individual components or in the final model, were combined when performing gene enrichment analysis. In order to better understand the contributions of individual gene features, we looked more closely at their weights in the final model of survival (Figure 6). We found that some of the binary clinical variables (ClinicalBin) successfully separated high- and low-risk patients in the STAD test set. Furthermore, the weights given to these features in the final composite model appeared reasonable. Being tumor free had a large negative weight, indicating decreased risk, while having a residual tumor had a positive weight, indicating increased risk. Similarly, having grade 2 or stage IB tumors or complete response to therapy were markers of good prognosis, while having grade 3 or stage IIIB tumors were markers of poor prognosis.

From the mutation allele format (MAF) dataset, we found that the most positive weight is from *ERBB3*, a member of the *EGFR* family that can heterodimerize with other family members, notably *ERBB2* [34]. Overexpression of *ERBB3* and *ERBB1* mRNAs and proteins in histological specimens of gastric cancer has been significantly associated with tumor relapse and poorly differentiated morphology [35]. From the RPPA data set, we observed that ERBB2 possesses a positive weight, and is therefore positively associated with worse outcome. This result, again, matches what is currently known in *ERBB2*-mutant gastric cancers and other cancers such as breast and lung cancers: overexpression of ERBB2 promotes tumor growth and survival by downstream signaling through the PI3K-AKT and RAS-MAPK pathways. Many of the results from the mRNAseq data set can also be validated from the literature. These include the fact that increased expression of *SYK* [36], *DMBT1* [37], *CASP7* [38], *OLFM4* [39], and *PIGR* [40] has been reported to be associated with prolonged survival. High expression of *MAGEA6* [41], *DKK1* [42], and the MYC-induced long noncoding RNA *H19* [43] has been associated with shorter survival.

In a wide variety of cancers, *TP53* mutations are generally associated with a poor prognosis, including in gastric adenocarcinomas when *TP53* was assessed by immunohistochemistry [44]. However, our analysis of the MAF data set found that the presence of a *TP53* mutation was associated with a negative weight, and therefore better overall survival. This result is supported by a univariate analysis using the STAD TCGA data set that reported that wild type *TP53* was associated with poorer prognosis in patients with gastric cancer [45], confirming independent findings reported previously [46,47,48]. However, several other studies found no association between *TP53* and survival in stomach cancer [49,50,51,52]. Earlier studies were characterized by a variety of methods to assess mutation status, including immunohistochemical staining [44] and limited direct sequence analysis of specific mutation hotspots [53]. Since our approach is based on whole exome sequencing of the entire gene, as well as the indirect contribution of genes such as *TP53* through various components, it is still possible that *TP53* has a small, non-independent effect that merely contributes to the effects of other mutations. We also found that *ATM* mutations and *ATM* expression were associated with a better outcome. Similar to mutated *TP53*, the literature contains conflicting results, with reports of low *ATM* expression associated with both poor [54] and good [55] prognosis. These apparently discrepant findings suggest that the association between mutation status and prognosis depends upon both the context and the methods used to assess mutation status.

The weights from the miRNA-seq data set have mixed agreement with what is known about the role of miRs in STAD. We found that miR-509 and miR-338 have negative weights and thus lower levels should be associated with worse prognosis, which agrees with the literature [56,57,58,59]. But we also found that miR-552, miR-105, and miR-196a have negative weights, yet the literature suggests that higher levels of these miRs are associated with tumor severity in gastric cancer or breast cancer [60,61,62]. Similarly, downregulation of miR-31 has been reported to be associated with worse overall survival of gastric cancer patients [63], which, again, is the opposite of our results. We are unable to fully explain the discrepancies between our findings and the literature. The differences may be related to the fact that many earlier studies used different technologies (for example, microarrays rather than sequencing). There may also be differences depending on whether assays are targeting primary miRNAs, precursor miRNAs, or 3′ or 5′ mature miRNAs.

## 5. Conclusions

We have identified a multiomic machine learning model that found composites of the variables, i.e., “components,” supervised with respect to patient outcome. We found 23 components, 14 of which were retained in the final PLASMA model. We validated the model with 1/3 of the STAD dataset and with the adenocarcinoma subset of the independent ESCA dataset. Using gene enrichment analysis, we were able to associate the predictive components with important pathways relevant to gastric cancer. By dissecting the weight matrix, we were also able to find a list of variables from different omics datasets with large contributions to patient outcome, many of which could be confirmed in the literature. Thus, we have demonstrated that PLASMA can be used as a discovery tool to highlight features associated with patient outcomes of interest. This process may ultimately prove useful by providing hints of novel targets in both gastric and other cancers.

## Figures and Tables

**Figure 1 cancers-17-00287-f001:**
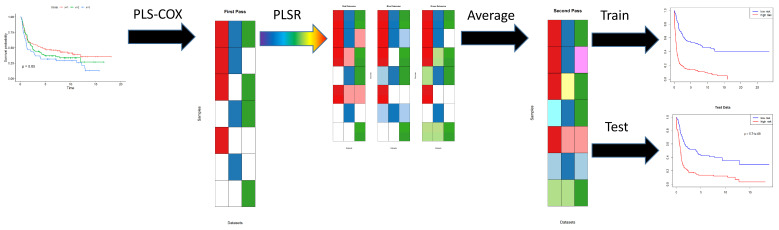
Workflow schematic for PLASMA algorithm with three omics data sets. Details of each step in the algorithm are explained in the main text.

**Figure 2 cancers-17-00287-f002:**
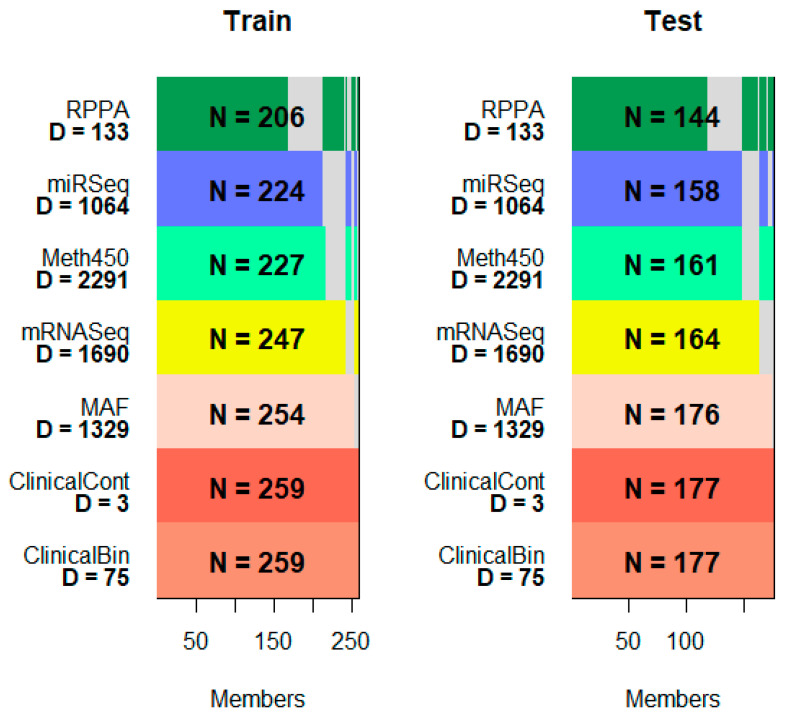
Overview of training and test data. (N is the number of samples in a data set; D is the number of features. RPPA = reverse phase protein arrays; Meth450 = Illumina 450K methylation arrays; MAF = mutation allele format; ClinicalBin = binary clinical features; ClinicalCont = continuous clinical features.).

**Figure 3 cancers-17-00287-f003:**
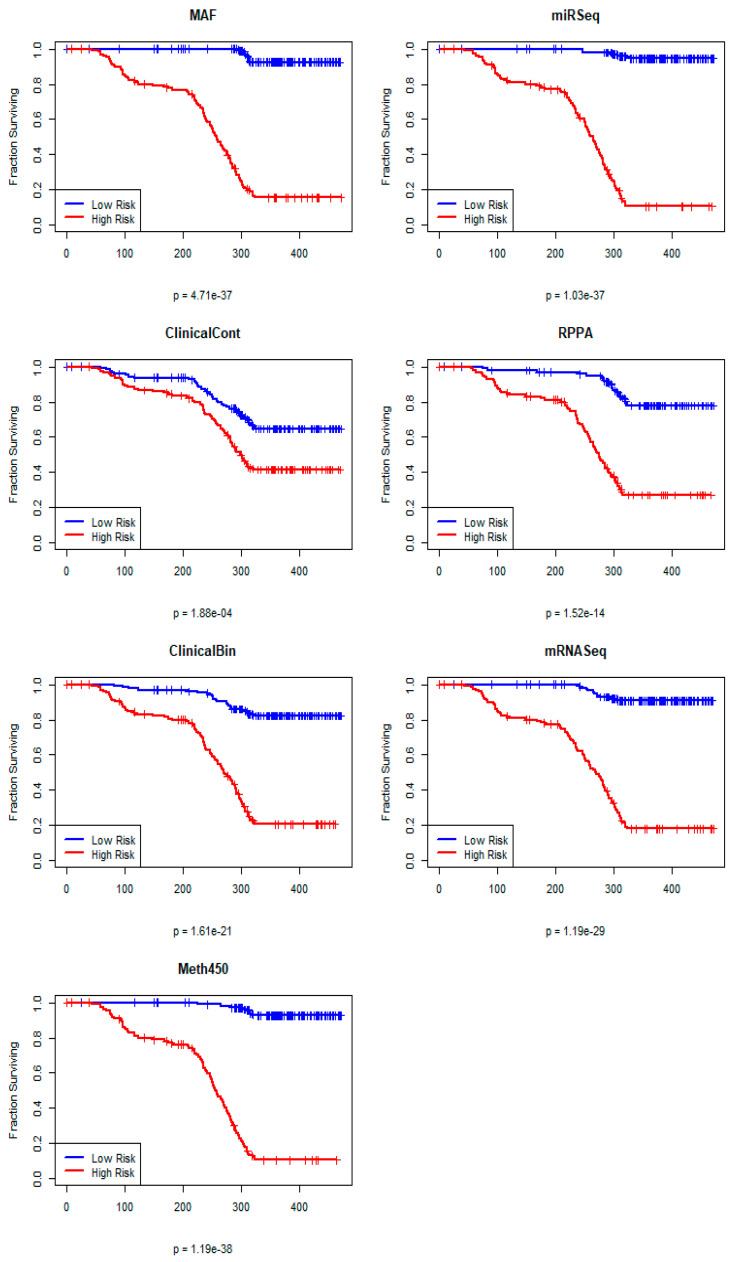
Kaplan−Meier plots of overall survival on the STAD training set from separate PLS Cox omics models.

**Figure 4 cancers-17-00287-f004:**
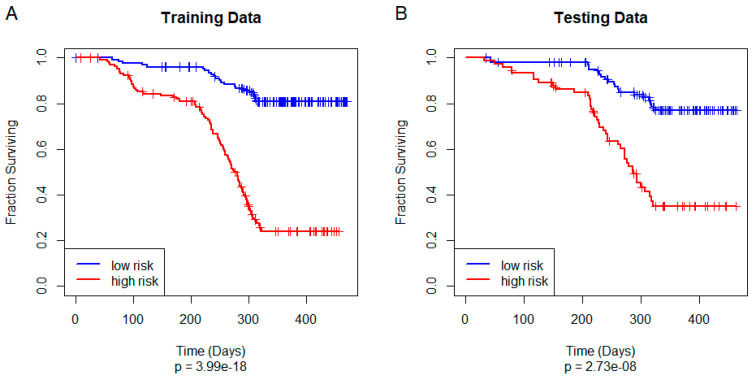
Kaplan−Meier plot of overall survival using the unified PLASMA Cox model on (**A**) the STAD training set and (**B**) the STAD test set.

**Figure 5 cancers-17-00287-f005:**
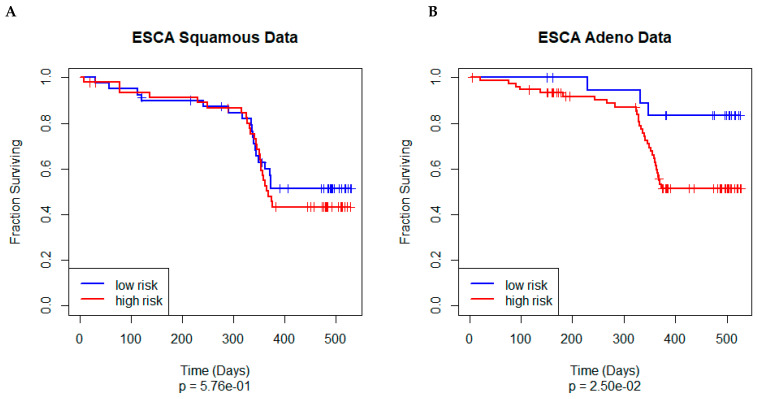
Kaplan−Meier plot of overall survival using the unified PLASMA Cox model on (**A**) the negative comparison ESCA squamous cell cancers and (**B**) the ESCA adenocarcinoma validation set.

**Figure 6 cancers-17-00287-f006:**
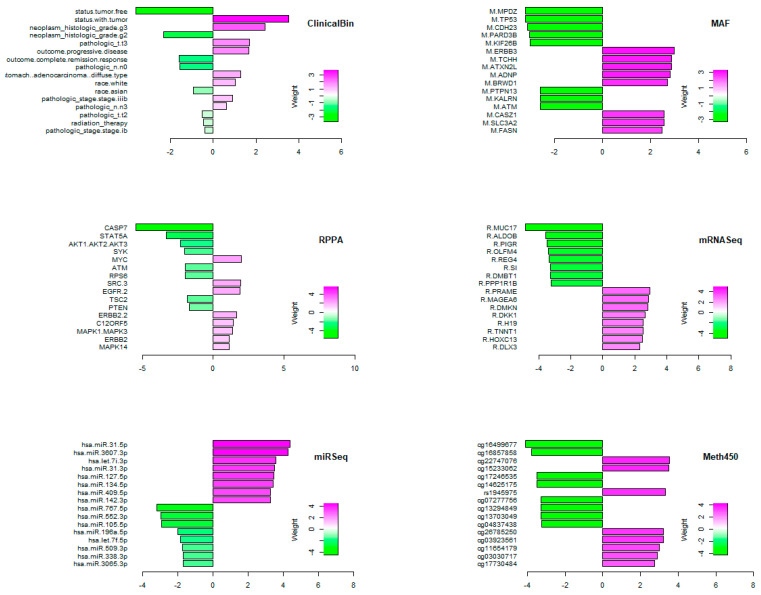
Highly weighted features in the final Cox model. All weights have been standardized within omics data set. Positive weights (magenta) indicate increased hazard ratio; negative weights (green) indicate decreased hazard ratio. Note that the ninth label on the “ClinicalBin” plot is left-truncated; the full version reads “histological.type.stomach..adenocarcinoma..diffuse.type”. Prefix M = MAF; prefix R = mRNASeq.

**Figure 7 cancers-17-00287-f007:**
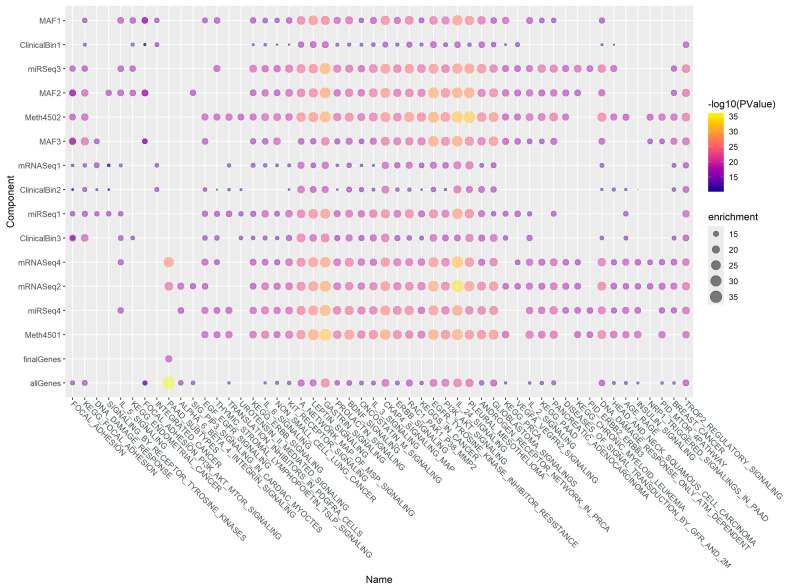
Clustering components based on pathway annotation *p*-values. Gene enrichment analysis was performed in ToppGene for individual and combined components. The Pearson correlation coefficient between *p*-values for enrichment of biological pathways was used to cluster both the components and the pathways. The sizes of bubbles reflect the enrichment score, and the colors reflect the *p*-value. The comma-separated-values (CSV) spreadsheet (Appendix A) maps the displayed values to the full original values.

**Table 1 cancers-17-00287-t001:** Coefficients of the omics components retained in the final model.

	coef	exp(coef)	se(coef)	z	*p*
MAF1	1.531	4.624	0.382	4.006	0.00006
MAF2	1.283	3.608	0.191	6.729	0.00000
MAF3	6.092	442.244	1.509	4.037	0.00005
miRSeq1	1.131	3.098	0.334	3.387	0.00071
miRSeq3	0.263	1.301	0.104	2.532	0.01135
miRSeq4	−0.985	0.373	0.449	−2.193	0.02833
ClinicalBin1	−1.700	0.183	0.955	−1.780	0.07501
ClinicalBin2	−6.407	0.002	3.309	−1.936	0.05285
ClinicalBin3	−4.426	0.012	1.961	−2.258	0.02397
mRNASeq1	−0.180	0.835	0.128	−1.405	0.16005
mRNASeq2	0.949	2.582	0.292	3.248	0.00116
mRNASeq4	0.779	2.178	0.436	1.788	0.07385
Meth4501	−0.095	0.909	0.041	−2.301	0.02140
Meth4502	−1.992	0.136	0.924	−2.156	0.03105

Abbreviations: coef = log hazard ratio; exp(coef) = hazard ratio; MAF = mutation allele frequency data component; miRSeq = microRNA sequencing data component; ClinicalBin = clinical binary data component; mRNASeq = mRNA sequencing data component; Meth = methylation data component.

## Data Availability

The results included here are wholly based upon data generated by the TCGA Research Network (https://www.cancer.gov/ccg/research/genome-sequencing/tcga). The plasma R package can be obtained from The Comprehensive R Archive Network (CRAN) at https://CRAN.R-project.org/package=plasma. The latest version of the package can always be obtained from R-Forge at https://r-forge.r-project.org/R/?group_id=1746. Source code and data for the full analysis presented here can be obtained from GitLab, at https://gitlab.com/krcoombes/plasma. All accessed on 5 December 2024.

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
