# Peer review of "PLASMA: Partial LeAst Squares for Multiomics Analysis"

_cancers, 2025, doi:10.3390/cancers17020287_

Round 1
Reviewer 1 Report
Comments and Suggestions for Authors
This manuscript presents an innovative algorithm, "plasma," that addresses the integration of multiomics datasets with incomplete data for supervised learning to predict time-to-event outcomes. The methodology is novel, the results are compelling, with strong performance in training, testing, and independent validation settings. I have some minor revision:
Minor Revision:
1. Some figures (e.g., heatmaps, bubble plots) require clearer legends and contextual explanations to enhance their interpretability.
2. The clustering results from gene enrichment analysis (e.g., Supplemental Figure S3) are very interesting. However, these data are not fully integrated into the main conclusion. How the data can help individualize biomarkes and/or pathways correlate to gastro-adenocarcinoma prognosis?
3. The name "plasma" is widely recognized in biomedical sciences as referring to patient-derived samples, blood plasma. This could create confusion among readers. My reccomandation is to either recast "plasma" as an acronym (e.g., P.L.A.S.M.A., Partial Least Squares Multiomics Analysis) or choose a new name that avoids overlap with biological terminology.
Author Response
This manuscript presents an innovative algorithm, "plasma," that addresses the integration of multiomics datasets with incomplete data for supervised learning to predict time-to-event outcomes. The methodology is novel, the results are compelling, with strong performance in training, testing, and independent validation settings.
Response: We thank the reviewer for their kind comments.
I have some minor revision:
Minor Revision:
1. Some figures (e.g., heatmaps, bubble plots) require clearer legends and contextual explanations to enhance their interpretability.
Response: We have expanded the captions for Figure 7 (bubble plot) and Supplemental Figure S3 (heatmap).
2. The clustering results from gene enrichment analysis (e.g., Supplemental Figure S3) are very interesting. However, these data are not fully integrated into the main conclusion. How the data can help individualize biomarkes and/or pathways correlate to gastro-adenocarcinoma prognosis?
Response: We have made no change in response to this comment. In preparing our initial draft, we debated internally whether to include (main) Figure 7 or Supplemental Figure S3 (or both) into the manuscript, since they are different graphical views of the same data. If the editor or reviewer prefers, we could swap the roles of these two figures (or include both). Moreover, the interpretation of these figures is already described in a paragraph in the discussion section (since the interpretation isn't a factual result).
3. The name "plasma" is widely recognized in biomedical sciences as referring to patient-derived samples, blood plasma. This could create confusion among readers. My reccomandation is to either recast "plasma" as an acronym (e.g., P.L.A.S.M.A., Partial Least Squares Multiomics Analysis) or choose a new name that avoids overlap with biological terminology.
Response: We have changed the name to use uppercase letters (PLASMA) instead of lowercase (plasma).
Reviewer 2 Report
Comments and Suggestions for Authors
The authors developed plasma algorithm for supervised prediction of survival using multiple datasets available, including multiomics datasets and clinical datasets. The idea of the algorithm is very much appreciated, and the factors identified from the model may be useful for future better prediction and study of cancers.
Major concerns:
1) Multiple datasets were used in the plasma algorithm, including RPPA, miRSeq, Meth450, mRNAseq, MAF (exome), ClinicalCont and ClincalBin. While the authors demonstrated the "unified" ("combined" seems better) prediction result was better than those from single "Omics" prediction, some components might have little contribution and could be eliminated as the authors pointed out. Therefore, I would like to see the comparison results after eliminating each dataset, so as to stand out the critical contributions from those imperative datasets.
2) The manuscript could be improved with better organization, with individual dataset training (2.2) and testing (2.4) together; combined datasets training and testing (2.3) with additional comparisons as mentioned aboved. Figure 2 changed into a table would be better. All the K-M plots could be smaller. And, Table 1 should be annotated and cited in the context.
3) It is difficult or awkward for me to identify pathways from ClinicalBin. Better interpretation is needed either in methods or context.
4) Please give reasons for the cutoffs used in the methods. e.g., Lines, 370, 374, 375, 381, 382, 445.
5) Is it possible to have a better prediction using the multiple factors that the authors have identified? Simplified model will be better.
Comments on the Quality of English Languagegrammar could be improved throughout the manuscript.
Author Response
The authors developed plasma algorithm for supervised prediction of survival using multiple datasets available, including multiomics datasets and clinical datasets. The idea of the algorithm is very much appreciated, and the factors identified from the model may be useful for future better prediction and study of cancers.
Response: We thank the reviewer for their kind comments.
Major concerns:
Comment 1) Multiple datasets were used in the plasma algorithm, including RPPA, miRSeq, Meth450, mRNAseq, MAF (exome), ClinicalCont and ClincalBin. While the authors demonstrated the "unified" ("combined" seems better) prediction result was better than those from single "Omics" prediction, some components might have little contribution and could be eliminated as the authors pointed out. Therefore, I would like to see the comparison results after eliminating each dataset, so as to stand out the critical contributions from those imperative datasets.
Response: First, the Merriam-Webster Dictionary definition of "combined" is "merged" or "united into a single number or expression". The Cambridge Dictionary, via its examples, adds the implication that it is close in meaning to "added", "summed" or "totaled". The Merriam-Webster definition of "unified" is "brought together as one". While the two terms are clearly related, we believe that "unified" more closely captures the meaning we wish to convey.
Second, we are not convinced that the additional analyses requested by the reviewer are actually necessary. The PLASMA algorithm already uses the Akaike Information Criterion (AIC) when it is building the final model that draws on all of the single omics analyses. The use of AIC automates the process of deciding which features, or even which entire data sets, are redundant or unnecessary in the final model.
Comment 2) The manuscript could be improved with better organization, with individual dataset training (2.2) and testing (2.4) together; combined datasets training and testing (2.3) with additional comparisons as mentioned above. Figure 2 changed into a table would be better. All the K-M plots could be smaller. And, Table 1 should be annotated and cited in the context.
Response: We have switched the order of sections 2.3 and 2.4, which accomplishes the reorganization requested. We particularly thank the reviewer for noticing that we somehow forgot to put an explicit reference to Table 1 into the manuscript. We have added a couple of sentences to rectify this problem. We have not converted Figure 2 into a table, but can easily do so if the editor prefers it that way. Nor have we changed the size of the Kapan-Meier plots; we would be happy to work with the copy editor to produce the final figures at an agreeable size.
Comment 3) It is difficult or awkward for me to identify pathways from ClinicalBin. Better interpretation is needed either in methods or context.
Response: We are not sure which part of the manuscript the reviewer is referring to. Our best guess is that this is the upper left panel of Figure 6, where the feature names are so long that the beginning is cut off in the plot labels. This difficulty originates in the source clinical data from TCGA, which does use quite long names in some places. We have shortened the names so that all but one of the features now fits on the plot, and added a comment in the text to explain the remaining term.
Comment 4) Please give reasons for the cutoffs used in the methods. e.g., Lines, 370, 374, 375, 381, 382, 445.
Response: The reasons for all the cutoffs were explained in the original draft at the beginning of the numbered list "We filtered the data sets so that only the most variable, and presumably the most informative, features were retained." In several cases, we created exploratory plots of the distributions of the means and standard deviations of the expression values or mutation rates to help choose the cutoffs. However, there was no formal statistical test applied to make the choice. In order for readers to reproduce our results, however, we felt it important to specify the exact cutoffs that we used.
Comment 5) Is it possible to have a better prediction using the multiple factors that the authors have identified? Simplified model will be better.
Response: We absolutely agree that a simplified model would be better. And we addressed that issue, to some extent, in the discussion, by pointing out that it is unlikely that clinical decisions will wait to incorporate a full multiomics analysis on each patient.
Round 2
Reviewer 2 Report
Comments and Suggestions for Authors
The authors claimed in the response that "The PLASMA algorithm already uses the Akaike Information Criterion (AIC) when it is building the final model that draws on all of the single omics analyses. The use of AIC automates the process of deciding which features, or even which entire data sets, are redundant or unnecessary in the final model." These should have been noted in some way in the context. Notably, abbreviation AIC in the manuscript has no full name, though in the supplementary materials.
Typo for "Conclusions" in the abstract is not acceptable.
Notes for Figure 6 in Page 8 needs certain revision.
Figures 2, 4, and 5 could be adjusted smaller, while Figure 6 could be adjusted larger for better view.
Author Response
Comment 1: The authors claimed in the response that "The PLASMA algorithm already uses the Akaike Information Criterion (AIC) when it is building the final model that draws on all of the single omics analyses. The use of AIC automates the process of deciding which features, or even which entire data sets, are redundant or unnecessary in the final model." These should have been noted in some way in the context. Notably, abbreviation AIC in the manuscript has no full name, though in the supplementary materials.
Response: The original version of the manuscript had the full name in Results Section 2.6. We have now added a sentence in Methods that spells out the full name again, and defines the acronym AIC.
Comment 2: Typo for "Conclusions" in the abstract is not acceptable.
Response: Fixed.
Comment 3: Notes for Figure 6 in Page 8 needs certain revision.
Response: We updated the caption by repeating the sentence that had been added in the main text.
Comment 4: Figures 2, 4, and 5 could be adjusted smaller, while Figure 6 could be adjusted larger for better view.
Response: We assume that the copy editor can adjust the sizes of the figures without any intervention on our part. If you would like us to make specific changes to the figures to assist with htis process, we would be happy to help.